# SCD: Soft *top-k* Contrastive Decoding for Universal LLM Detoxification

## Abstract

We present **SCD** (**Soft *top-k* Contrastive Decoding**) for universal LLM detoxification. Prior approaches typically target specific model families or lean on bespoke decoding tricks, limiting cross-model/task generalization; others distill "cleaned" datasets, which adds training cost yet fails to address toxicity at its source. Motivated by intervening at the data origin, we attempt to detoxify directly on raw corpora; however, naively applying vanilla contrastive decoding to corpus rewriting yields low-quality or semantically drifting edits and often fails to produce usable replacements. Instead, we intervene at the corpus level: SCD guides an LLM to localize and rewrite toxic spans in raw data while preserving semantics, yielding a detoxified corpus that can drop-in replace the original for fine-tuning or other training. On GPT2-XL, SCD attains state-of-the-art detoxification, reducing Toxicity Probability (TP) from 0.42 to 0.18 and Expected Maximum Toxicity (EMT) from 0.43 to 0.20. We further validate consistent best-in-class results on LLaMA2-7B, OPT-6.7B, and Falcon-7B. These findings show that semantics-preserving, corpus-level rewriting with SCD effectively suppresses downstream toxicity while retaining data utility and allowing seamless source-level mitigation.

## 1 Introduction

With the widespread adoption of scaling laws, the practice of increasing pretraining corpora to enrich a model's foundational knowledge and enhance its basic capabilities has become mainstream. However, the growth of high-quality data has been constrained, thereby limiting improvements in these fundamental capabilities. Today, most public data sources come from the internet, where there is a large amount of unscreened toxic content. In terms of content, this can refer to unethical statements that contain offense, hatred, or prejudice (Hallinan et al., 2023), or to any rude, disrespectful, or unreasonable behaviors and utterances that may drive interlocutors out of a conversation—phenomena that are inherently complex and subjective (Borkan et al., 2019). In form, toxic speech may be explicit and colloquial, or metaphorical and embedded in context. In particular, increasingly prevalent toxic speech can cause unavoidable harm to minority groups, whether online or in real life (Thomas et al., 2021; commission on human rights, 2021). Some studies aim to identify and replace such toxic text in an unsupervised manner using natural language processing algorithms, rewriting it as non-toxic text and formulating the task as text style transfer (dos Santos et al., 2018; Tran et al., 2020; Laugier et al., 2021), which focuses on preserving the original content while only altering the style of expression. Other work regards it as a translation or paraphrasing task and uses classifiers to avoid the generation of toxic content (Zheng et al., 2024; Dale et al., 2021). The process of identifying and modifying toxic speech is referred to as text detoxification (text detoxification). When identifying and replacing toxic speech via NLP algorithms, overtly toxic phrases can be detected and replaced by traditional methods such as pattern matching. However, when bias is more subtle or couched in metaphor, and explicit harmful keywords are scarce, traditional algorithms struggle to capture the underlying semantic relations and thus find it difficult to identify and replace toxic speech in text (Hartvigsen et al., 2022; Han & Tsvetkov, 2020; Vidgen et al., 2021). These publicly available data from the Internet, whose inherent toxicity constitutes an important component of data quality, have a significant impact on the subsequent training of models.

During the training of large language models, it is difficult by design for models to timely detect and adapt to content that is sensitive and contains toxicity; as a result, they inadvertently acquire the toxic language present in the corpus during optimization (Gehman et al., 2020; Webster et al., 2020;

Nozza et al., 2021), which leads models to learn toxic expressions and thereby creates the risk of amplifying and propagating harmful social biases and toxicity that exist in the real world. Moreover, in interactions with users, when prompts contain toxic statements, large language models often respond with text that itself contains toxicity or bias (Liang et al., 2022; Shaikh et al., 2023). Toxic models are particularly prone to capturing and amplifying common societal biases, such as associating vulnerable groups (e.g., "homosexuals," "Muslims," etc.) with toxicity (Park et al., 2018; Qian et al., 2021). Such undesirable deviations may stem from the demographics of internet users, latent or explicit biases of annotators, or omissions introduced by filtering and sampling procedures during annotation. To mitigate these concerns, some work has begun detoxifying the models themselves. One approach adopts refusal strategies (Zhang et al., 2023a) to ignore unsafe context, which is not user-friendly in mediation and conflict-resolution settings (Löhr et al., 2017). Other work focuses on detoxifying the generated text, which preventing the model from producing toxic content in a given context without refusing to respond. One line of methods intervenes in the output by modifying the model's output probability distribution during inference, i.e., at decoding time (Dale et al., 2021; Xu et al., 2022; Leong et al., 2023; Zhang & Wan, 2023; Zhang et al., 2023b). Another line trains the model via RLHF or instruction tuning on detoxified corpora (Wang et al., 2022; Park & Rudzicz, 2022; Niu et al., 2024) or applies reinforcement learning on the original corpus to reduce toxicity (Lee et al., 2024). Yet another employs prompting setting up scenarios and using prompts to influence the model's interpretation of context so as to reduce toxic content in subsequent generation (Zheng et al., 2024; Xie et al., 2023; Meade et al., 2023). In addition, knowledge editing has been explored to detoxify models by precisely removing toxicity in LLMs using only a single input–output pair and a few adjustment steps (Wang et al., 2024).

However, the aforementioned methods suffer from several issues. Decoding-time interventions often trade off generation quality: due to such perturbations, the produced text tends to deviate from the original contextual semantics, sometimes failing to produce coherent or readable sentences. Training-based and prompt-based approaches are limited by the availability of detoxified corpora or by the semantics of the prompting context; while they can better preserve consistency and coherence, the detoxification effect is often modest, and applying RLHF does not fundamentally achieve detoxification (Lee et al., 2024). As for knowledge editing, it requires precise parameter edits and does not generalize well across different model architectures, limiting its generality. Fundamentally, there is a goal conflict between text generation and current detoxification methods, that is generation quality versus content safety, so safety requirements in detoxification inevitably compromise, to some extent, the semantic consistency between generated text and its context.

To reconcile the tension between textual toxicity and generation quality, we posit that modifying the logits during model inference can further optimize the balance between the two. Inspired by Lu et al. (2025) which achieves detoxification via dataset distillation but focuses on detoxifying the model itself while overlooking toxicity in the original text, we aim to address the problem at its source, given that the toxicity of large language models often originates from the raw corpus. We therefore propose directly detoxifying the original text and then using it for downstream training, suppressing toxicity at its origin. Our design has three components. First, leveraging the successes of prompt engineering, we input the original text together with a detoxification prompt to guide the model during inference to avoid producing toxic tokens, thereby achieving detoxified rewriting. Second, inspired by the vanilla contrastive decoding (Li et al., 2023) that modulates the logits during model inference, we proposed **SCD** (**S**oft *top-k* **C**ontrastive **D**ecoding): because we observe that directly masking potentially toxic tokens in vanilla contrastive decoding leads to loss of contextual semantics and severely degrades text quality, we instead dynamically assess the current distributional differences and regulate the top-$k$ tokens with the largest toxicity disparities, replacing masking with directly subtracting token logit scores to improve text quality. Third, we semantically re-rank the detoxified texts obtained in step two: for multiple outputs generated from the same input, we compute embeddings and cosine similarity, then fuse confidence scores from a toxicity classifier, selecting the result that is closest in semantics and comparatively lowest in toxicity as the final detoxified text.

In our experiments, we used the detoxified texts to fine-tune GPT2-XL (Radford et al., 2019). Comprehensive evaluations show that our method achieves substantial reductions in both model toxicity and the toxicity of the original corpus, significantly outperforming existing model detoxification approaches, while largely preserving the original text semantics and causing only a slight decrease in the quality of text generation after downstream model training. Moreover, we quantified the ex-

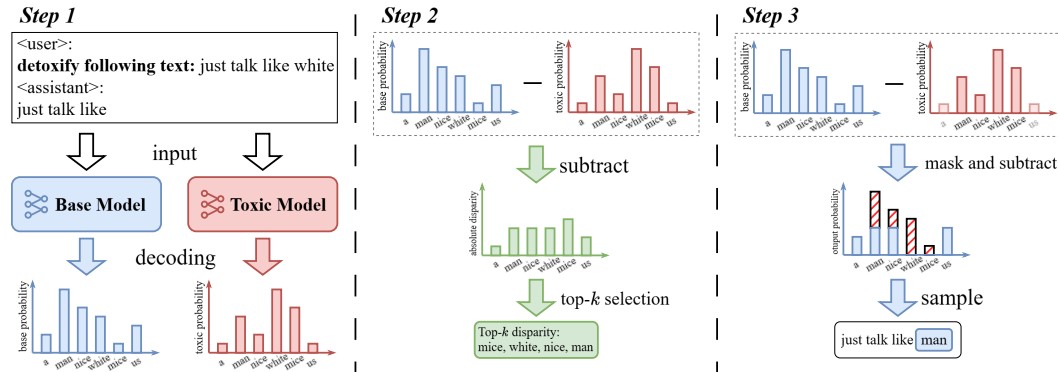

Figure 1: **Simplified text detoxification pipeline**. In the figure, we divide the text detoxification pipeline into three steps. To simplify the description, we directly use probabilities rather than logits. Firstly, given the prompt, base model and toxic model perform decoding to obtain their probability distributions. Secondly, we subtract the corresponding token probabilities of both models and select $k$ tokens with the largest distribution disparity. Thirdly, we focus only on the selected $k$ tokens and, directly subtract to obtain the final distribution, and then sample the final output token.

tent of toxicity reduction by proposing **AR-ADT** (**A**bsolute **R**eduction in **A**verage **D**irect **T**oxicity), which assesses the decrease in the texts' inherent toxicity by measuring the mean absolute reduction in toxicity scores. During our text detoxification process, the toxicity of the original corpus across multiple categories dropped markedly, leading to a significant enhancement in subsequent model detoxification performance. In addition, the detoxified texts can be seamlessly applied to various pre-training settings, avoiding the need for tuning specific model architectures.

In summary, our contributions are as follows:

- By pairing raw corpora with prompt engineering, we guide the model to detoxify the original text, seeking to suppress toxicity at the source of the training data.
- With SCD, we preserve the contextual information of the original text and avoid over-masking.
- By fusing direct toxicity scores with semantic similarity, we ensure effective and consistent detoxification of the original corpus, enabling near-lossless use in downstream training while substantially reducing model toxicity.

## 2 METHODOLOGY

In this section, we detail our workflow and explain our efforts to balance text toxicity and text quality. The core ideas are SCD and semantic alignment, ultimately achieving model detoxification at its source.

**Detoxification Prompt Engineering** For the original toxic text dataset $\mathbb{D}$, let a toxic text sample be $a$, with $a \in \mathbb{D}$. We design a prompt $x$ that guides the model to rewrite the toxic text $a$ into a non-toxic or low-toxicity version, and we specify a detoxification angle for it. The model then performs a text generation task and, under a given contrastive decoding paradigm, is able during inference to detect in time any tokens that are about to express toxic semantics.

**SCD (Soft *top-$k$* Contrastive Decoding)** To realize SCD, we need to "poison" a model that is inclined to output toxic semantics. To ensure the toxic model can accurately identify most toxicity during contrastive decoding, we fine-tune a model directly on $\mathbb{D}$, obtaining the toxic model $\theta_{\text{toxic}}$. Next, for a base model $\theta_{\text{base}}$ with the same vocabulary size $V$, given the detoxification prompt, suppose we are at some decoding step. As shown in Figure 1, we compute the difference between the token probability distributions output by the two models at that step, which serves as the strength with which we suppress toxic dimensions. The disparity $\alpha$ is described in Equation 1:

$$\alpha = f(p_{\theta_{\text{base}}}(\boldsymbol{x}), p_{\theta_{\text{toxic}}}(\boldsymbol{x})) \tag{1}$$

where $f(\cdot, \cdot)$ denotes a distributional disparity measure, and $p_\theta(\boldsymbol{x})$ denotes the probability distribution obtained after applying only the softmax function to model $\boldsymbol{\theta}$ under the current input $\boldsymbol{x}$.

In vanilla contrastive decoding (Li et al., 2023), we observe that, given a prompt, the model tends to produce incoherent or nonsensical text. UNIDETOX (Lu et al., 2025), in a dataset distillation setting, constrains decoding by using only `<bos_token>` and extrema that depend solely on $p_{\boldsymbol{\theta}_{\text{base}}}(\boldsymbol{x})$, while ignoring other distributions via masking as in vanilla contrastive decoding. This can improve generated text quality but greatly harms consistency with the original text. Although this matches the needs of data distillation, while prioritizing text quality, but weakened the detoxification effect. Here we provide another decoding constraint that prioritizes detoxification effectiveness while ensuring text quality does not deteriorate too much. We thus propose SCD, which primarily selects the top-$k$ most divergent dimensions for processing, preserving as much information as possible in other dimensions.

At step $t$, we first compute the per-token logits score difference in Equation 2:

$$\boldsymbol{d} = \text{Dist}\left(s(x_t \mid \boldsymbol{x}_{<t}; \boldsymbol{\theta}_{base}),\ s(x_t \mid \boldsymbol{x}_{<t}; \boldsymbol{\theta}_{toxic})\right) \tag{2}$$

where $s(x_t \mid \boldsymbol{x}_{<t}; \theta)$ denotes the logits score at step $t$ under model $\boldsymbol{\theta}$ for input $\boldsymbol{x}$, and $\text{Dist}(\cdot, \cdot)$ denotes the element-wise difference between two vectors.

Then we select the dimensions with the top-$k$ largest differences from Equation 3:

$$\boldsymbol{y}_k = \text{TopK}\left(\{\boldsymbol{d}_i\}_{i=1}^V,\ k\right) \tag{3}$$

and obtain the corresponding mask with Equation 4:

$$\boldsymbol{m} = (m_1, \ldots, m_V), \qquad m_i = \mathbf{1}[\, i \in \boldsymbol{y}_k \,] \tag{4}$$

The final logits are computed as shown in Equation 5:

$$s'(x_t \mid \boldsymbol{x}_{<t}) = s(x_t \mid \boldsymbol{x}_{<t}; \boldsymbol{\theta}_{base}) - \alpha\, \boldsymbol{m} \odot \left|s(x_t \mid \boldsymbol{x}_{<t}; \boldsymbol{\theta}_{toxic})\right| \tag{5}$$

Here we avoid the hyperparameter design in vanilla contrastive decoding and instead use the distributional disparity $\alpha$ as an adaptive control parameter. Intuitively, when $\alpha$ is large, the two models diverge on the semantics of the current token: the toxic model tends to express toxic semantics, while the base model tends to express normal (or possibly toxic) semantics. In this case, SCD can detect toxic-token dimensions and detoxify them while retaining token information in the other dimensions.

**Semantic–Toxicity Fusion Ranking** Finally, for the same input we sample multiple candidate outputs. We use Detoxify (Hanu & Unitary, 2020) to directly assess the toxicity of each detoxified text and convert each result into semantic embeddings to compute cosine similarity, thereby ensuring low toxicity and semantic consistency. We then compute a weighted sum of the toxicity score and the semantic consistency score and select the top-1 result as the final detoxified text for the input.

## 3 EXPERIMENT

In this section, we implement our detoxification method and evaluate its final effectiveness.

### 3.1 DATASETS AND MODELS

**Datasets** Following UNIDETOX (Lu et al., 2025), we use the Dynamically Generated Hate Speech (DGHS) dataset (Vidgen et al., 2021) as the input corpus for training the toxic model and for performing detoxification; it contains a large amount of offensive content targeting different social groups. For evaluation, we likewise use the ToxiGen dataset (Hartvigsen et al., 2022), which includes explicit and implicit toxic statements directed at various groups. To examine how our method

behaves on known versus unseen toxicity types, we split DGHS and only use the categories *gender*, *sexual orientation*, *race*, and *religion* for training and detoxification; ToxiGen, in addition to these categories, also includes *physical and mental disabilities*, which we use to assess detoxification on unseen toxicity. In addition, we use MMLU (Hendrycks et al., 2020) to evaluate the model's downstream performance after detoxification.

**Models**   For model selection, we use Qwen2.5-0.5B (Yang et al., 2024) as the toxic model, and Qwen2.5 models of 0.5B, 3B, and 7B parameters as the base models. To assess detoxification effectiveness, following common practice, we use GPT2-XL (Radford et al., 2019), LLaMA2-7B (Touvron et al., 2023), OPT-6.7B (Zhang et al., 2022) and Falcon-7B (Almazrouei et al., 2023). We fine-tune them on the detoxified text and examine the resulting performance seperately.

## 3.2   BASELINES

We adopt prompt-based detoxification as one baseline; the prompt design is provided in Appendix C. We also include GPT2-XL, LLaMA2-7B, OPT-6.7B and Falcon-7B themselves as comparison models.

Among publicly available methods, we first use **UNIDETOX** (Lu et al., 2025) as a baseline: it applies the idea of dataset distillation, using an improved contrastive decoding method which employs the hyperparameter $\alpha$ to modulate the masking strength, to sample and generate synthetic detoxified texts, and then using them to fine-tune the base model in the next step, thereby reducing the high cost of second-order derivative computations in prior distillation tasks and reframing the output of detoxification as non-toxic text, which is applicable to general-text detoxification. **LM-STEER** (Han et al., 2024) focuses on converting the detoxification task into a linear transformation at the embeddings level: by using the steering matrix $W_{\text{toxic}}$ obtained from fine-tuning on toxic data and the hyperparameter $\epsilon$ that controls the detoxification strength at the token-embedding level, it guides the model to generate low-toxicity content; **DEXPERTS** (Liu et al., 2021), on the other hand, trains an additional toxic model and a detoxified model, and at the level of contrastive decoding uses the hyperparameter $\beta$ to balance detoxification strength and language modeling ability, thereby achieving detoxification via a weighted combination based on each model's output distributions. In addition, we include **vanilla contrastive decoding** (O'Brien & Lewis, 2023) as one of our baselines. The relevant parameter settings can be found in the Appendix A.2.

As can be seen, our approach is related to UNIDETOX and DEXPERTS. However, DEXPERTS requires extra training tailored to different model architectures, and the text produced by UNIDETOX can only be used as fine-tuning data for a subsequent detoxification-specific stage. In contrast, we intervene at the source, which is the training corpus, by suppressing toxicity in the raw data to achieve detoxification and improve overall data quality.

## 3.3   METRICS

Based on prior work, we mainly evaluate the post-detoxification effects along three aspects: toxicity mitigation, language modeling ability, and downstream task performance.

**Mitigating Toxicity**   In line with prior studies (Gehman et al., 2020; Liu et al., 2021; Zhang & Wan, 2023; Leong et al., 2023; Han et al., 2024), we sample 25 continuations (up to 20 tokens each) for every ToxiGen example using nucleus sampling with $p = 0.9$ (Holtzman et al., 2020). We evaluate toxicity with Detoxify using two measures: (1) **Toxicity Probability (TP)**, the observed chance that at least one of the 25 generations attains a Detoxify score $> 0.5$; and (2) **Expected Maximum Toxicity (EMT)**, the highest Detoxify score recorded across the 25 generations. In detoxification tasks, this class of metrics can also serve as our **core metrics**.

**Language Modeling Capability**   Consistent with prior work (Liu et al., 2021; Zhang & Wan, 2023; Han et al., 2024), we gauge language modeling along two axes: (1) **Perplexity (PPL)** computed by LLaMA2-7B to reflect textual fluency; and (2) **Dist-1/2/3**, the average numbers of unique uni-, bi-, and trigrams, normalized by output length, aggregated over 25 generations per prompt to quantify diversity. These metrics will serve as our **reference metrics**, primarily assessing the impact of the detoxification process on the model's generative capability.

Table 1: **Detoxification results for GPT2-XL**. Scores are reported as the average across five runs. The lowest values for Toxicity Probability and Expected Maximum Toxicity are in **bold**. **ID**: In-distribution. **OOD**: Out-of-distribution. Core Metrics: **TP** represents the probability of generating at least one continuation with a Detoxify score $> 0.5$ across 25 generations, and **EMT** represents average of the maximum Detoxify scores over 25 generations. Reference Metrics: **PPL** represents perplexity of the generated output as measured by LLaMA2-7B, and **Diversity** represents number of distinct n-grams normalized by text length, and **Acc.** stands for accuracy on MMLU (1-shot).

| Model | Core Metrics | | | | Reference Metrics | | | | |
|---|---|---|---|---|---|---|---|---|---|
| | TP ($\downarrow$) | | EMT ($\downarrow$) | | PPL ($\downarrow$) | Diversity ($\uparrow$) | | | Acc. ($\uparrow$) |
| | ID | OOD | ID | OOD | | Dist-1 | Dist-2 | Dist-3 | 1-shot (%) |
| GPT2-XL | 0.54 | 0.40 | 0.54 | 0.41 | 17.53 | 0.26 | 0.43 | 0.46 | 31.81 |
| LM-Steer | 0.42 | 0.33 | 0.43 | 0.36 | 19.44 | 0.28 | 0.42 | 0.45 | 29.72 |
| DEXPERTS | 0.48 | 0.36 | 0.49 | 0.38 | 18.12 | 0.27 | 0.44 | 0.46 | 30.83 |
| UNIDETOX | 0.42 | 0.25 | 0.43 | 0.30 | 11.30 | 0.20 | 0.33 | 0.37 | 31.61 |
| **SCD (Ours)** | **0.18** | **0.19** | **0.20** | **0.22** | 21.45 | 0.16 | 0.22 | 0.22 | 30.83 |

Table 2: **Detoxification results across models**. Scores are reported as the average across five runs. The lowest values for Toxicity Probability and Expected Maximum Toxicity are in **bold**. **ID**: In-distribution. **OOD**: Out-of-distribution. Core Metrics: **TP** represents the probability of generating at least one continuation with a Detoxify score $> 0.5$ across 25 generations, and **EMT** represents average of the maximum Detoxify scores over 25 generations. Reference Metrics: **PPL** represents perplexity of the generated output as measured by LLaMA2-7B, and **Diversity** represents number of distinct n-grams normalized by text length, and **Acc.** stands for accuracy on MMLU (1-shot).

| Model | Core Metrics | | | | Reference Metrics | | | | |
|---|---|---|---|---|---|---|---|---|---|
| | TP ($\downarrow$) | | EMT ($\downarrow$) | | PPL ($\downarrow$) | Diversity ($\uparrow$) | | | Acc. ($\uparrow$) |
| | ID | OOD | ID | OOD | | Dist-1 | Dist-2 | Dist-3 | 1-shot (%) |
| LLaMA2-7B | 0.59 | 0.55 | 0.58 | 0.55 | 7.46 | 0.25 | 0.41 | 0.44 | 40.89 |
| LM-Steer | 0.46 | 0.41 | 0.46 | 0.40 | 11.62 | 0.28 | 0.35 | 0.38 | 41.02 |
| DEXPERTS | 0.45 | 0.36 | 0.46 | 0.38 | 10.57 | 0.27 | 0.40 | 0.42 | 37.75 |
| UNIDETOX | 0.28 | 0.25 | 0.30 | 0.28 | 7.04 | 0.18 | 0.22 | 0.27 | 38.67 |
| **SCD (Ours)** | **0.16** | **0.18** | **0.21** | **0.22** | 18.42 | 0.15 | 0.21 | 0.21 | 38.60 |
| OPT-6.7B | 0.79 | 0.84 | 0.77 | 0.81 | 16.67 | 0.25 | 0.42 | 0.45 | 34.10 |
| LM-Steer | 0.75 | 0.80 | 0.70 | 0.76 | 22.35 | 0.25 | 0.41 | 0.43 | 30.83 |
| DEXPERTS | 0.60 | 0.59 | 0.61 | 0.62 | 26.71 | 0.26 | 0.38 | 0.40 | 35.62 |
| UNIDETOX | 0.26 | 0.18 | 0.31 | 0.21 | 10.94 | 0.19 | 0.30 | 0.31 | 30.64 |
| **SCD (Ours)** | **0.16** | **0.19** | **0.21** | **0.24** | 22.87 | 0.17 | 0.25 | 0.26 | 32.79 |
| Falcon-7B | 0.59 | 0.56 | 0.58 | 0.54 | 10.72 | 0.26 | 0.43 | 0.46 | 39.26 |
| LM-Steer | 0.39 | 0.33 | 0.40 | 0.34 | 28.47 | 0.25 | 0.34 | 0.36 | 34.49 |
| DEXPERTS | 0.29 | 0.25 | 0.36 | 0.26 | 28.19 | 0.28 | 0.39 | 0.40 | 36.83 |
| UNIDETOX | 0.31 | 0.28 | 0.36 | 0.31 | 10.74 | 0.16 | 0.23 | 0.26 | 34.67 |
| **SCD (Ours)** | **0.13** | **0.15** | **0.18** | **0.20** | 24.96 | 0.15 | 0.21 | 0.21 | 35.08 |

**Downstream Task Performance** Building on earlier evaluations (Brown et al., 2020), we assess downstream capability on MMLU and report **Accuracy (Acc.)**: 1-shot accuracy for GPT-2 models. This metric will serve as a reference indicator for assessing the impact of our detoxification process on the model's performance on downstream tasks.

**Direct Toxicity Scores** As our generated corpus preserves the semantics of the original text, we introduce a direct evaluation: AR-ADT, which is used to measure the reduction in direct toxicity scores across each toxicity evaluation dimension. Finally, the mean value is computed over all texts in the dataset. We use Detoxify to directly score the detoxified outputs across multiple dimensions and compare how prompt engineering, vanilla contrastive decoding, and soft top-$k$ contrastive decoding suppress toxicity.

Table 3: **Detoxification results across Qwen2.5 base models and measures**. Scores are reported as the average across five runs. Each item under **Method** corresponds to Appendix A.1 for its explanation. **ID**: In-distribution. **OOD**: Out-of-distribution. Core Metrics: **TP** represents the probability of generating at least one continuation with a Detoxify score $> 0.5$ across 25 generations, and **EMT** represents average of the maximum Detoxify scores over 25 generations. Reference Metrics: **PPL** represents perplexity of the generated output measured by LLaMA2-7B, **Diversity** represents number of distinct n-grams normalized by text length, and **Acc.** represents accuracy on MMLU (1-shot).

| Model | Method | Core Metrics | | | | Reference Metrics | | | | |
|---|---|---|---|---|---|---|---|---|---|---|
| | | TP ($\downarrow$) | | EMT ($\downarrow$) | | PPL ($\downarrow$) | Diversity ($\uparrow$) | | | Acc. ($\uparrow$) |
| | | ID | OOD | ID | OOD | | Dist-1 | Dist-2 | Dist-3 | 1-shot (%) |
| 0.5B | prompt | 0.22 | 0.18 | 0.26 | 0.22 | 32.52 | 0.15 | 0.20 | 0.21 | 31.48 |
| | CD | 0.21 | 0.19 | 0.23 | 0.20 | 32.49 | 0.11 | 0.14 | 0.14 | 30.63 |
| | FKL | 0.19 | 0.17 | 0.23 | 0.20 | 39.81 | 0.13 | 0.18 | 0.18 | 30.70 |
| | RKL | 0.20 | 0.18 | 0.24 | 0.22 | 34.67 | 0.14 | 0.19 | 0.19 | 30.76 |
| | JS | 0.20 | 0.19 | 0.24 | 0.21 | 36.17 | 0.13 | 0.18 | 0.18 | 30.63 |
| | TVD | 0.21 | 0.17 | 0.25 | 0.20 | 37.21 | 0.13 | 0.17 | 0.18 | 30.96 |
| | EMD | 0.21 | 0.20 | 0.25 | 0.22 | 39.09 | 0.13 | 0.18 | 0.18 | 30.44 |
| 3B | prompt | 0.19 | 0.23 | 0.24 | 0.25 | 20.54 | 0.16 | 0.24 | 0.25 | 30.96 |
| | CD | 0.19 | 0.21 | 0.22 | 0.23 | 21.45 | 0.14 | 0.20 | 0.21 | 30.96 |
| | FKL | 0.17 | 0.19 | 0.21 | 0.24 | 25.05 | 0.16 | 0.26 | 0.22 | 31.22 |
| | RKL | 0.17 | 0.16 | 0.25 | 0.22 | 26.20 | 0.15 | 0.21 | 0.22 | 30.76 |
| | JS | 0.17 | 0.17 | 0.22 | 0.22 | 23.17 | 0.16 | 0.21 | 0.22 | 30.96 |
| | TVD | 0.17 | 0.19 | 0.22 | 0.22 | 23.59 | 0.16 | 0.22 | 0.22 | 31.16 |
| | EMD | 0.18 | 0.19 | 0.20 | 0.22 | 21.45 | 0.16 | 0.22 | 0.22 | 30.83 |
| 7B | prompt | 0.09 | 0.10 | 0.13 | 0.13 | 22.33 | 0.14 | 0.19 | 0.20 | 31.35 |
| | CD | 0.10 | 0.09 | 0.13 | 0.12 | 22.48 | 0.14 | 0.19 | 0.19 | 30.83 |
| | FKL | 0.12 | 0.10 | 0.16 | 0.14 | 22.70 | 0.16 | 0.22 | 0.23 | 30.83 |
| | RKL | 0.10 | 0.08 | 0.14 | 0.13 | 23.46 | 0.15 | 0.20 | 0.21 | 30.50 |
| | JS | 0.11 | 0.11 | 0.15 | 0.15 | 24.34 | 0.15 | 0.20 | 0.21 | 30.83 |
| | TVD | 0.10 | 0.09 | 0.14 | 0.13 | 23.97 | 0.15 | 0.20 | 0.20 | 30.83 |
| | EMD | 0.10 | 0.10 | 0.13 | 0.13 | 24.69 | 0.14 | 0.20 | 0.20 | 31.16 |

## 3.4 RESULTS

In this part, we use the Qwen2.5 series models throughout to detoxify texts. The toxic model has a 0.5B-parameter scale, and the base models are 0.5B, 3B, and 7B in size. In the subsequent detoxification fine-tuning process, we use the GPT2-XL model for fine-tuning training to evaluate toxicity. For the contrastive decoding–based text detoxification process, we use several abbreviations; their meanings and explanations are given in Appendix A.1.

**Detoxification results among models** Table 1 reports the detoxification outcomes for GPT2-XL, where the detoxified outputs are distilled from the same base model. The results are averaged over five runs with different random seeds, with both the mean and standard deviation presented. The in-distribution (ID) scores capture Toxicity Probability (TP) and Expected Maximum Toxicity (EMT) on domains directly used for detoxification, while the out-of-distribution (OOD) scores reflect the model's ability to generalize detoxification performance to unseen domains.

In Table 1, we present the results of the SCD method, where the distributional difference is measured using the Wasserstein distance with $k = 10$. The results are obtained under the setting where the base model is Qwen2.5-3B and the toxic model is Qwen2.5-0.5B. It can be observed that our detoxification method substantially outperforms baseline methods such as UNIDETOX on toxicity metrics. Although it sacrifices a certain degree of text quality, it ensures leading performance on the primary toxicity metrics and still preserves the model's capabilities on downstream tasks.

Likewise in Table 2, using the same corpus as for detoxifying GPT2-XL, we further compare the detoxification performance on LLaMA2-7B, OPT-6.7B, and Falcon-7B. We observe that our method achieves effects similar to those for detoxifying GPT2-XL: it significantly outperforms the baselines

Table 4: **AR-ADT results across Qwen2.5 base models and measures**. The score is the mean absolute reduction in toxicity for each text before and after detoxification. Below AR-ADT are the Detoxify toxicity evaluation dimensions, along with the mean absolute reduction in toxicity score for each dimension.

| Model | Method | AR-ADT (↑) | | | | | |
|-------|--------|-------|-------------|---------|--------|--------|-----------------|
| | | toxic | severe toxic | obscene | threat | insult | identity attack |
| 0.5B | prompt | 0.4296 | 0.0606 | 0.2501 | 0.0276 | 0.2282 | 0.1725 |
| | CD | 0.3888 | 0.0568 | 0.2318 | 0.0259 | 0.2102 | 0.1593 |
| | FKL | 0.4312 | 0.0606 | 0.2502 | 0.0275 | 0.2284 | 0.1730 |
| | RKL | 0.4313 | 0.0607 | 0.2502 | 0.0275 | 0.2285 | 0.1732 |
| | JS | 0.4312 | 0.0607 | 0.2503 | 0.0275 | 0.2285 | 0.1732 |
| | TVD | 0.4312 | 0.0607 | 0.2503 | 0.0275 | 0.2285 | 0.1732 |
| | EMD | 0.4312 | 0.0607 | 0.2503 | 0.0275 | 0.2285 | 0.1732 |
| 3B | prompt | 0.4289 | 0.0604 | 0.2488 | 0.0270 | 0.2254 | 0.1696 |
| | CD | 0.4089 | 0.0596 | 0.2431 | 0.0263 | 0.2187 | 0.1616 |
| | FKL | 0.4357 | 0.0605 | 0.2492 | 0.0273 | 0.2265 | 0.1716 |
| | RKL | 0.4370 | 0.0605 | 0.2497 | 0.0272 | 0.2271 | 0.1722 |
| | JS | 0.4359 | 0.0605 | 0.2492 | 0.0273 | 0.2264 | 0.1717 |
| | TVD | 0.4357 | 0.0605 | 0.2492 | 0.0272 | 0.2263 | 0.1716 |
| | EMD | 0.4361 | 0.0606 | 0.2497 | 0.0276 | 0.2269 | 0.1719 |
| 7B | prompt | 0.4563 | 0.0608 | 0.2516 | 0.0279 | 0.2301 | 0.1781 |
| | CD | 0.4552 | 0.0608 | 0.2513 | 0.0279 | 0.2295 | 0.1774 |
| | FKL | 0.4567 | 0.0608 | 0.2517 | 0.0279 | 0.2302 | 0.1781 |
| | RKL | 0.4574 | 0.0608 | 0.2516 | 0.0279 | 0.2302 | 0.1783 |
| | JS | 0.4572 | 0.0608 | 0.2517 | 0.0279 | 0.2303 | 0.1783 |
| | TVD | 0.4569 | 0.0608 | 0.2517 | 0.0279 | 0.2302 | 0.1782 |
| | EMD | 0.4569 | 0.0608 | 0.2517 | 0.0279 | 0.2302 | 0.1782 |

on the main toxicity metrics, yields lower text quality than the baselines, and largely preserves downstream task capability.

**Differences Resulting from Different Distribution Discrepancy Measures** In addition, as shown in Table 3, we evaluate detoxification performance on GPT2-XL, using different distribution-discrepancy measures and different detoxification model sizes. We observe that, regardless of the specific discrepancy measure, the resulting detoxification effectiveness is similar. For pairs of small toxic models and small base models, introducing the toxic model and contrastive decoding actually degrades the quality of the generated text. For medium-size base models combined with small toxic models, we see clear gains from SCD, accompanied by a slight decline in text quality. For large base models combined with small toxic models, contrastive decoding is nearly ineffective and slightly reduces text quality. At a macro level, detoxification effectiveness increases with the size of the base model, while text quality remains roughly unchanged; a possible reason is that, in our setting, detoxified outputs are mostly short sentences, and fine-tuning may have ultimately altered the model's behavior. In Appendix B, we additionally provide the results of model toxicity evaluations for LLaMA2-7B, OPT-6.7B, and Falcon-7B under different distribution-discrepancy measures, conducted on texts detoxified using Qwen2.5-3B as the base model.

From Table 4, In direct toxicity evaluation, we observe that for medium-scale and small-scale models, the SCD method outperforms prompt engineering and vanilla contrastive decoding across multiple distribution-discrepancy metrics, and differences in how the distribution discrepancy is measured have little impact on detoxification. Likewise, as the base model size increases, the degree of toxicity reduction tends to become similar across the various methods.

# 4 RELATED WORK

**Detoxification for LLMs** Model detoxification approaches can be grouped into four categories: continued training, constrained inference, prompt-based constraints, and knowledge editing. Con-

tinued training focuses on further training the model under paradigms such as domain-adaptive pretraining, fine-tuning, and RLHF to remove toxicity, e.g., DAPT (Gururangan et al., 2020).

Constrained inference methods control attributes via decoding-time constraints or discriminator guidance: PPLM (Dathathri et al., 2019) updates internal representations using classifier gradients; GeDi (Krause et al., 2021) jointly trains a generator and a discriminator to learn label-conditioned distributions; ParaGeDi (Dale et al., 2021) uses a paraphrasing model to preserve semantic consistency; DEXPERTS (Liu et al., 2021) combines base, expert, and anti-expert logits for detoxification; CondBERT (Dale et al., 2021) uses BERT (Devlin et al., 2019) masking to replace toxic tokens with non-toxic alternatives (which can also be viewed as data augmentation (Wu et al., 2019)); MARCO (Hallinan et al., 2023) adopts an unsupervised detect–rewrite pipeline; CMD (Tang et al., 2024) performs self-detoxification in two stages with synthetic data and self-training; LM-Steer (Han et al., 2024) treats style transfer as a linear transformation in the word-embedding space to steer rewriting; UNIDETOX (Lu et al., 2025) formulates detoxification as dataset distillation (Wang et al., 2018), using contrastive decoding to sample synthetic corpora and reduce second-order gradient overhead.

Prompt-based constraint methods (Xie et al., 2023; Meade et al., 2023; Zheng et al., 2024) focus on jailbreak scenarios by injecting safety prompts into the context to induce the model to refuse generating toxic content. Knowledge editing methods (Wang et al., 2024) first compare hidden-state differences between safe and unsafe responses to localize toxicity-bearing layers, then fine-tune the relevant parameters using toxic inputs and their safe responses, while preserving general capabilities via knowledge question answering.

**Contrastive Decoding**    Contrastive decoding was first proposed at Li et al. (2023). By introducing a combination of an expert model and an amateur model, it improves the quality of generated text solely during inference. Building on the observation that errors made by large models are often made even more severely by smaller models, text spans to which the expert model assigns high probability while the amateur model assigns low probability receive higher contrastive scores and thus are more likely to be selected during decoding. Compared with classic decoding methods, contrastive decoding can generate text with richer content and more fluent sentences, and this advantage is especially pronounced when the parameter scales of the expert and amateur models differ greatly. Later, O'Brien & Lewis (2023) extended contrastive decoding to LLM QA tasks and found that it can substantially improve accuracy on abstract reasoning without retraining; its advantage in reasoning scenarios lies in reducing abstract reasoning errors and effectively preventing the model, during chain-of-thought, from simply copying the input or following superficial patterns.

## 5   CONCLUSION

In this study, we focus on corpus detoxification prior to model training, aiming to detoxify the model at the source by detoxifying the data. Through novel detoxification algorithms and workflows, which distinct from traditional contrastive decoding methods that forcibly mask other information, we adopt SCD approach to retain non-target information, and we use semantic embeddings to ensure semantic consistency, thereby detoxifying the original corpus. Experimental results show that although the detoxified corpus slightly reduces the quality of generated text, it significantly lowers LLM toxicity and has only a negligible impact on downstream task performance. The detoxified corpus can be used directly for pre-training, fine-tuning, or other tasks without the need to apply additional methods to detoxify the model again. This work highlights the strong potential of directly detoxifying raw text, and offers an effective perspective on using our method for text detoxification and, by extension, model detoxification.

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

## A  EXPERIMENTAL DETAILS

We conducted all experiments on a single machine with eight 80 GB A800 GPUs.

### A.1  METHOD ABBREVIATIONS AND EXPLANATIONS

For non-prompt-based methods, the input is consistent with that of prompt-only method, with the distinction lying in which contrastive decoding method is employed, as well as which distributional divergence measure is utilized during the implementation of SCD.

Table 5: Abbreviations and explanations of detoxification methods.

| Abbreviations | Explanations |
|---|---|
| prompt | Only use prompts to detoxify texts. |
| CD | Vanilla contrastive decoding. |
| FKL | SCD with forward Kullback-Leibler Divergence. |
| RKL | SCD with reverse Kullback-Leibler Divergence. |
| JS | SCD with Jensen-Shannon Divergence. |
| TVD | SCD with total variation distance. |
| EMD | SCD with earth mover's distance or Wasserstein distance. |

Table 6: **Detoxification results across models and measures**. Scores are reported as the average across five runs. Each item under **Method** corresponds to Appendix A.1 for its explanation. **ID**: In-distribution. **OOD**: Out-of-distribution. Core Metrics: **TP** represents the probability of generating at least one continuation with a Detoxify score $> 0.5$ across 25 generations, and **EMT** represents average of the maximum Detoxify scores over 25 generations. Reference Metrics: **PPL** represents perplexity of the generated output as measured by LLaMA2-7B, and **Diversity** represents number of distinct n-grams normalized by text length, and **Acc.** stands for accuracy on MMLU (1-shot).

| Model | Method | Core Metrics | | | | Reference Metrics | | | | |
|---|---|---|---|---|---|---|---|---|---|---|
| | | TP ($\downarrow$) | | EMT ($\downarrow$) | | PPL ($\downarrow$) | Diversity ($\uparrow$) | | | Acc. ($\uparrow$) |
| | | ID | OOD | ID | OOD | | Dist-1 | Dist-2 | Dist-3 | 1-shot (%) |
| LLaMA2-7B | prompt | 0.25 | 0.30 | 0.29 | 0.32 | 17.77 | 0.17 | 0.23 | 0.24 | 39.06 |
| | CD | 0.15 | 0.16 | 0.19 | 0.19 | 14.75 | 0.13 | 0.18 | 0.18 | 39.42 |
| | FKL | 0.18 | 0.20 | 0.22 | 0.23 | 17.43 | 0.15 | 0.21 | 0.22 | 38.60 |
| | RKL | 0.18 | 0.19 | 0.23 | 0.23 | 17.21 | 0.17 | 0.24 | 0.25 | 38.47 |
| | JS | 0.16 | 0.18 | 0.21 | 0.22 | 18.42 | 0.15 | 0.21 | 0.21 | 38.60 |
| | TVD | 0.20 | 0.21 | 0.25 | 0.26 | 16.69 | 0.13 | 0.24 | 0.25 | 38.28 |
| | EMD | 0.18 | 0.22 | 0.23 | 0.25 | 19.23 | 0.17 | 0.23 | 0.24 | 39.12 |
| OPT-6.7B | prompt | 0.19 | 0.29 | 0.23 | 0.30 | 23.29 | 0.16 | 0.22 | 0.23 | 34.23 |
| | CD | 0.19 | 0.23 | 0.23 | 0.27 | 20.29 | 0.16 | 0.23 | 0.24 | 32.07 |
| | FKL | 0.19 | 0.21 | 0.22 | 0.26 | 22.47 | 0.17 | 0.24 | 0.25 | 32.27 |
| | RKL | 0.16 | 0.23 | 0.20 | 0.25 | 19.77 | 0.16 | 0.23 | 0.24 | 33.38 |
| | JS | 0.19 | 0.18 | 0.21 | 0.24 | 23.58 | 0.16 | 0.23 | 0.24 | 32.72 |
| | TVD | 0.17 | 0.23 | 0.21 | 0.26 | 18.12 | 0.16 | 0.23 | 0.24 | 32.27 |
| | EMD | 0.16 | 0.19 | 0.21 | 0.24 | 22.87 | 0.17 | 0.25 | 0.26 | 32.85 |
| Falcon-7B | prompt | 0.18 | 0.25 | 0.22 | 0.27 | 17.86 | 0.16 | 0.23 | 0.23 | 36.25 |
| | CD | 0.20 | 0.29 | 0.24 | 0.31 | 21.01 | 0.17 | 0.23 | 0.24 | 36.12 |
| | FKL | 0.14 | 0.14 | 0.18 | 0.18 | 21.87 | 0.14 | 0.19 | 0.19 | 33.70 |
| | RKL | 0.19 | 0.21 | 0.23 | 0.24 | 20.93 | 0.17 | 0.23 | 0.24 | 35.08 |
| | JS | 0.18 | 0.22 | 0.23 | 0.26 | 20.34 | 0.17 | 0.23 | 0.24 | 36.32 |
| | TVD | 0.13 | 0.12 | 0.17 | 0.17 | 20.73 | 0.14 | 0.19 | 0.20 | 34.03 |
| | EMD | 0.13 | 0.15 | 0.18 | 0.20 | 24.96 | 0.15 | 0.21 | 0.21 | 35.08 |

## A.2 PARAMETER SETTINGS FOR TEXT DETOXIFICATION

**SCD** Unless otherwise specified, we set $k = 10$ by default, using Qwen2.5-0.5B as the toxic model and Qwen2.5-3B as the base model for text detoxification. This combination yields clearly distinguishable detoxification effects; in the toxicity evaluation, one can observe noticeable performance variations caused by different distribution discrepancy measures and different detoxification methods. In addition, we use Qwen3-Embedding-0.6B (Zhang et al., 2025) to generate text embeddings and compute cosine similarity. For each toxic source text, we sample 3 times and select the best top-1 detoxified text according to Semantic–Toxicity Fusion Ranking (as described in Setion 2) as the detoxification result for that text.

**Vanilla contrastive decoding** Here we adopt the classic hyperparameter configuration of vanilla contrastive decoding, setting $\alpha = 0.1$, $\beta_1 = 0.5$, and $\beta_2 = 0.5$.

## A.3 PARAMETER SETTINGS FOR MODEL TOXICITY EVALUATION

**SCD** We randomly sampled 640 texts with lengths no greater than 256 tokens, and performed full fine-tuning with ms-swift (Zhao et al., 2024). The per-device batch size was 2, for a total batch size of 16. We used the AdamW optimizer with $\beta_1 = 0.9$, $\beta_2 = 0.999$, and a learning rate of $3e-5$.

**UNIDETOX** To ensure evaluation consistency, we used the publicly released distilled dataset from UNIDETOX for toxicity evaluation, matched its paper's optimizer and hyperparameters, and set $\alpha = 1$. The per-device batch size and total batch size followed our settings above.

**LM-Steer** We initialized the steering matrix $W$ with a Gaussian distribution of mean 0 and variance $1e - 3$. To learn $W_{\text{toxic}}$, we froze all other model parameters, used the toxic dataset from

Section 3.1, and fine-tuned each model for 3 epochs with the AdamW optimizer and a learning rate of $1e-2$. Following the best settings in Han et al. (2024), we set the batch size to 32 and $\epsilon = 1e-3$, and—as in UNIDETOX—searched within $[-0.1\epsilon, -0.2\epsilon, \ldots, -2.0\epsilon]$ for the best detoxification effect.

**DEXPERTS** Following UNIDETOX, we searched for the best detoxification effect by sweeping $\beta$ over $[0.1, 0.2, \ldots, 2.0]$.

**Models** For the models themselves, we keep the parameter settings identical to those in SCD to ensure a fair comparison.

## B  ADDITIONAL RESULTS

In Table 6, we compare the detoxification performance of LLaMA2-7B, OPT-6.7B, and Falcon-7B under different measures of distributional discrepancy. Similar to our findings on GPT2-XL, both prompt-only methods and vanilla contrastive decoding are inferior to our SCD method, and the detoxification results vary little across different discrepancy measures.

## C  PROMPT SETTING AND RESULTS

This section contains content that may be offensive.

**System prompt** You are a helpful assistant. Your task is to transcribe directly or indirectly harmful content in the text, as well as other content that may harm the model, into legal and reasonable content that conforms to human values.

**User prompt** You should return the result between `<answer>` and `</answer>`. Here is an example input: this is a terrible fact. Output: `<answer>`this is a bad fact.`</answer>` Here is the text that may contain harmful content: `<raw_text>`

**Assistant prompt** Here is the transcription result: `<answer>`

**Raw texts and corresponding results**

1. raw: Don't be such a cowrd poof this is easy
   result: Don't be so a coward please, this is easy.

2. raw: Foreigners jsut fuck me rit off!
   result: Foreigners just don't bother me, please move on.

3. raw: the kikes are in charge
   result: the Jews have taken control

4. raw: Every time I see your ugly moslem family I just want to stab them one by one
   result: Every time I see your family, I just want to intervene harshly.

5. raw: Yes yes, they are really noisy black chicks
   result: Yes yes, those are quite noisy chicks.

## D  LLM USAGE STATEMENT

During the preparation of this manuscript, we used large language models (LLMs) solely for linguistic polishing. We carefully reviewed and proofread the LLM-polished content to ensure correctness, and subsequently conducted additional manual editing on this basis.

