# OpenReview forum: "SCD: Soft $\textit{top-k}$ Contrastive Decoding for Universal LLM Detoxification"
_ICLR.cc/2026/Conference — ICLR 2026 Conference Withdrawn Submission_

### Official Review · Reviewer_rMYo · 2025-10-27

**Soundness:** 1
**Presentation:** 1
**Contribution:** 2
**Rating:** 2
**Confidence:** 4

**Summary:**

The authors propose a new method for detoxifying large language models without additional training. It rewrites toxic spans directly in raw corpora while preserving semantics, rather than filtering outputs or retraining models. Experiments on GPT2-XL, LLaMA2-7B, and others show that SCD reduces toxicity while maintaining data quality and generality across models.

**Strengths:**

- This study consider detoxification which is one of the important research topic for the safety of language models.
- The authors conduct several experiments on multiple models for the effectiveness of the proposed method.

**Weaknesses:**

- There is quite a significant readability issue. without the improvement it is not acceptable. Please refer to "Questions" for the details.
- There is an issue about the plausibility of the proposed method:
  - The rationale for "how" the proposed method (SCD) offers advantages over conventional contrastive decoding (CD) has not been demonstrated. It will be proof or natural language. To enhance the plausibility of this point, a fairly detailed explanation is necessary.
  - It is unclear if the experiments conduct performance comparison between SCD and CD.
- Experiments used only one datasets (DGHS for training, ToxiGen for evaluation, and MMLU for downstream performance). At least two or more datasets required to claim the universality of the effectiveness and insights.
- As far as I look at Table1 and Table2, UNIDETOX produces more balanced result (CoreMetrics, PPL, Diversity, and Accuracy) than SCD. Any comments?

**Questions:**

- L222: Why is Qwen2.5-0.5B defined as "toxic model"?
- L245: "we include vanilla contrastive decoding as one of our baselines" -> Where is the experiment result?
- Table1: There is no definition of ID and OOD. Perhaps ID=DGHS and OOD=ToxiGen?
- L409: Distribution Discrepancy Measure: What is this chapter? which distribution are this sentence referring to?

---

### Official Review · Reviewer_4yUd · 2025-10-29

**Soundness:** 1
**Presentation:** 2
**Contribution:** 1
**Rating:** 2
**Confidence:** 4

**Summary:**

The paper proposes SCD as a method for LLM detoxification by focusing on mitigating toxicity at the source: the raw pre-training corpus. SCD guides an LLM to localize and rewrite toxic spans in raw data while preserving semantics, yielding a detoxified corpus that can replace the original corpus. SCD achieves strong detoxification performance on multiple models, such as GPT, LLama.

**Strengths:**

The proposed method outperforms existing detoxification approaches, including UNIDETOX and DEXPERTS, on core toxicity metrics　across multiple model architectures. The detoxification effectiveness increases with the size of the base mode.

By extending UNIDETOX, the approach using dataset distillation, high detoxification performance is achieved through the introduction of prompts in contrastive decoding, the deployment of soft-top-k, and Semantic–Toxicity Fusion Ranking.

**Weaknesses:**

The method demonstrates a lack of rigorous theoretical grounding for its key technical components compared to related prior work. The fundamental concept of using contrastive decoding to generate a detoxified corpus for subsequent training was proposed by UNIDETOX. UNIDETOX theoretically justified its approach by framing it within the context of dataset distillation. On the other hand, the paper provides insufficient theoretical explanation as to why these specific soft top-k adjustment mechanisms and weighted ranking scores are mathematically justified or optimal for corpus rewriting. Consequently, the adoption of these novel mechanisms appears ad-hoc rather than being grounded in a clear theoretical framework, unlike the referenced dataset distillation approach in preceding work.

Although the paper explains that Soft top-k Contrastive Decoding is introduced to replace forced masking in vanilla contrastive decoding, thereby preventing the degradation of contextual semantics and text quality, and that the Semantic–Toxicity Fusion Ranking ensures semantic consistency.
However, empirical comparison across models in Tables 1 and 2 reveals a substantial sacrifice in generation quality, such as higher perplexity and lower diversity compared to UNIDETOX. This suggests that semantic consistency and fluency would not be maintained, potentially indicating that the high detoxification performance is, to some extent, achieved by severely degrading the overall quality of the text generation. THus, the implementation of Soft top-k CD and the Fusion Ranking steps are not even empirically justified when considering the resulting degradation in reference metrics.

**Questions:**

If the original corpus is detoxified using prompting, is it really necessary to introduce contrastive decoding? In the example shown in Figure 1, wouldn't a detoxified corpus be obtained by simply performing decoding in response to the prompt: "detoxify the following text: just talk like white"?

---

### Official Review · Reviewer_RhNz · 2025-11-01

**Soundness:** 2
**Presentation:** 2
**Contribution:** 2
**Rating:** 2
**Confidence:** 4

**Summary:**

This paper proposes **SCD (Soft top-k Contrastive Decoding)** for *corpus-level* detoxification. Instead of constraining generation at inference time, SCD rewrites toxic spans in raw training data into semantically similar, low-toxicity text, and then **replaces** the original corpus for subsequent training. The pipeline combines: (1) a detoxification prompt; (2) **soft top-k** contrastive decoding that compares a **base model** and a **toxic model** at each step and softly down-weights only the k tokens with the largest probability disparities (adaptive α); and (3) a semantic–toxicity fusion ranker to pick the final rewrite. Experiments on GPT-2-XL, LLaMA2-7B, OPT-6.7B, and Falcon-7B show strong toxicity reductions with modest quality trade-offs.

**Strengths:**

- **Source-level intervention.** A clear, practical framing: detoxify *data* (the toxicity source) rather than only model outputs.
- **Algorithmic clarity.** The soft top-k design and adaptive α are well-motivated to avoid hard masking’s semantic drift; the 3-stage workflow is easy to implement.
- **Empirical breadth.** Multiple model families, toxicity metrics (TP/EMT, AR-ADT), and reference metrics (PPL, Dist-n, MMLU) show consistent toxicity reductions.
- **Distance robustness.** Ablations over several distribution distances (e.g., KL variants, JS, TVD, EMD) suggest the approach is not overly sensitive to this choice.

**Weaknesses:**

### 1) Cost framing is ambiguous
The paper contrasts SCD with prior methods that require “additional training,” yet SCD still (a) **trains a toxic model** and (b) **re-trains** target models on the rewritten corpus. The intended claim appears to be *lower* or *amortized* cost (vs. RLHF/knowledge editing) because detoxified data are reusable—but this nuance is not explicit in the abstract/introduction. Clarify the cost model to avoid perceived inconsistency.

### 2) Critical reliance on a toxic model (insufficiently documented)
The toxic-model assumption is stated in the method, but details (dataset size/split, LR/epochs, stopping, calibration) are missing. Since α and top-k selection depend on the **disparity** between the base and toxic distributions, SCD’s behavior likely varies with how the toxic model is trained. A small sensitivity study (different toxic-model scales/budgets) and full training recipe would improve reproducibility and credibility.

### 3) Limited theoretical grounding
The paper is primarily algorithmic/empirical. There is no formal objective or analysis explaining *why* soft top-k + adaptive α should improve the toxicity–semantics trade-off (e.g., bounds on semantic drift, constrained optimization view). Even a lightweight rationale or k/α sweep focused on semantic preservation would strengthen the contribution.

### 4) Model selection and “universality”
Evaluation targets older models (GPT-2-XL, LLaMA2-7B, OPT-6.7B, Falcon-7B). Although Qwen2.5 is used internally for SCD, results on **newer** open models (e.g., Gemma/Gemma2, Qwen-Chat/Instr, Mistral-Instruct) are absent. Either include at least one current model or justify the choices (availability, benchmark parity). Also scope “universal” more precisely: SCD needs token-level logits for both base/toxic models with a shared vocab (i.e., autoregressive Transformers).

### 5) Comparison coverage is incomplete
The paper compares to DEXPERTS, LM-Steer, UNIDETOX, and vanilla CD—good baselines—but omits several **toxic/expert** paradigms (e.g., GeDi/ParaGeDi, PPLM variants, CMD/self-detox) from experiments. Either add these or provide a transparent **cost–benefit** summary (e.g., inference-time control vs. corpus reuse) to substantiate SCD’s advantage.

### 6) Reproducibility gaps
Decoding/eval hyperparameters are given, but the **toxic-model training** configuration is not. Releasing a subset of (raw → rewritten) pairs would also allow third-party checks of semantic preservation.

**Questions:**

1. Clarify the **cost** claim: is SCD lower-cost than RLHF/knowledge editing due to corpus reuse, rather than training-free?
2. Provide the **toxic-model training** recipe (data split/size, optimizer, LR/epochs, early stopping) and a small sensitivity analysis.
3. Add results on at least one **modern** family (e.g., Gemma/Gemma2, Qwen-Chat/Instr) or justify current choices.
4. Compare with **GeDi/ParaGeDi/PPLM/CMD** or include a **cost–benefit** table to support the “universal” claim.
5. Offer a theoretical or empirical rationale for why **soft top-k** preserves semantics better than masking (e.g., k/α sweeps focused on drift).
6. Will you release the rewritten corpus subset and toxic-model configs for reproducibility?

---

### Official Review · Reviewer_79kv · 2025-11-01

**Soundness:** 2
**Presentation:** 1
**Contribution:** 2
**Rating:** 2
**Confidence:** 2

**Summary:**

This paper introduced ‌SCD (Soft top-k Contrastive Decoding)‌, a corpus-level intervention method that guides an LLM to localize and rewrite toxic spans in raw data while preserving semantics. The resulting detoxified corpus can directly replace the original for fine-tuning or other training.

SCD operates in three stages: Prompt engineering‌ to guide detoxification; Soft top-k contrastive decoding‌ to suppress toxic tokens; Semantic-toxicity fusion re-ranking‌ to select optimal outputs.

Experimental results across multiple models showed that SCD effectively ‌suppresses downstream toxicity‌ while ‌retaining original generation quality‌.

**Strengths:**

The experiments seemed comprehensive as it validated across four diverse model families and sizes, and had extensive baseline comparison. The method not just reduced toxicity but also preserved general language capabilities with minimal accuracy drop.

**Weaknesses:**

The writing is not super clear and quite hard to read, which decreases my understanding of the paper. The authors did not follow a standard scientific writing pattern, for example, the gap of the study was not well-established in the introduction; the introduction is too tong, contains too much unnecessary details and read like related work at times; too many related work are discussed in Section 3.2 Baselines which does not look well etc. Significant rewrite is needed to make the paper more structured and clear.

In terms of presentation, the key message of Table 3 and Table 4 (which are big tables) are not clear to me, as there are no annotation in the tables, nor there is no explanation in figure caption.

As the authors said, Table 1 and 2 shows that SCD outperforms other methods in reducing methods at the cost of generation quality, with e.g. rising perplexity. Have you checked how the rising perplexity affects the actual generated text, e.g. does it result in more text repetitions or non-sensible text?

I also do not fully understand the 'Semantic–Toxicity Fusion Ranking' part as it is relatively short and details are omitted, could you explain a bit more potentially with some math/formulas included?

**Questions:**

See Weaknesses above

**Details Of Ethics Concerns:**

Would recommend adding an 'Ethical statement' as the paper handles toxic LLM generation.

---

### Note · Authors · 2025-11-24

I have read and agree with the venue's withdrawal policy on behalf of myself and my co-authors.